# Predictive Value of Neutrophil–Lymphocyte Ratio as a Marker in Antiresorptive Agent-Related Osteonecrosis of the Jaw: A Retrospective Analysis

**DOI:** 10.3390/diagnostics12081836

**Published:** 2022-07-29

**Authors:** Kazuto Kurohara, Kasumi Shimizu, Taku Murata, Gaku Koizumi, Akira Takigawa, Kokoro Nagata, Kenya Okumura, Naoya Arai

**Affiliations:** Department of Oral and Maxillofacial Surgery, Mie University Graduate School of Medicine, 2-174 Edobashi, Tsu 514-8507, Japan; skasumi@med.mie-u.ac.jp (K.S.); muratat@med.mie-u.ac.jp (T.M.); k-gaku@med.mie-u.ac.jp (G.K.); d-takira@med.mie-u.ac.jp (A.T.); kokoro-m@med.mie-u.ac.jp (K.N.); okuken@med.mie-u.ac.jp (K.O.); n-arai@med.mie-u.ac.jp (N.A.)

**Keywords:** neutrophil–lymphocyte ratio, platelet–lymphocyte ratio, osteonecrosis, cancer, osteoporosis, inflammation, antiresorptive agent-related osteonecrosis of the jaw

## Abstract

Antiresorptive agent-related osteonecrosis of the jaw (ARONJ), a multifactorial disease, can drastically affect a patient’s quality of life. Moreover, disease progression to severe acute inflammation can hinder treatment. Therefore, we aimed to investigate the diagnostic value of the neutrophil–lymphocyte ratio (NLR) and platelet–lymphocyte ratio (PLR) in predicting the risk of acute inflammation in patients with ARONJ. In total, 147 patients with ARONJ were enrolled between 1 January 2011 and 31 December 2019. They were divided into two groups according to their baseline NLR (high NLR vs. low NLR) or PLR (high PLR vs. low PLR) to analyze the relationship between NLR and PLR and the outcomes of acute inflammatory events. An optimal NLR cut-off value of 2.83 was identified for hospitalization for an inflammatory event. Logistic regression analysis showed that NLR > 2.83 was associated with an increased risk of hospitalization for an inflammatory event. A PLR cut-off value of 165.2 was identified for hospitalization for an inflammatory event. However, logistic regression analysis showed that PLR > 165.2 was not significantly associated with hospitalization for an inflammatory event. Our study findings suggest that the NLR has diagnostic value in predicting the risk of hospitalization for inflammatory events among patients with ARONJ.

## 1. Introduction

Bone-modifying agents (BMAs), such as bisphosphonates, are very useful in the treatment of metastatic bone cancer, osteoporosis, and autoimmune diseases. However, BMA use is associated with serious side effects. The administration of intravenous bisphosphonates, such as zoledronic acid, may lead to the development of bisphosphonate-related osteonecrosis of the jaw [1,2,3], acute systemic inflammatory reaction, ocular inflammation, renal failure, nephrotic syndrome, and electrolyte imbalance [4]. In addition, drugs like denosumab can cause side effects, such as necrosis of the jawbone. The terms antiresorptive agent-related osteonecrosis of the jaw (ARONJ) and medication-related osteonecrosis of the jaw were coined to describe these conditions. ARONJ is defined as exposure of necrotic bone in the maxillofacial region that lasts for >8 weeks, without a history of radiation treatment [1]. Progression of ARONJ to acute inflammation results in a marked decline in the patient’s quality of life (QOL). The presence of acute inflammation also interferes with the treatment of the main disease. However, there is no available index for predicting the development of the acute symptoms of ARONJ. The neutrophil–lymphocyte ratio (NLR) and platelet–lymphocyte ratio (PLR) have been investigated as markers of the systemic inflammatory response in several tumors [5,6,7,8] and other diseases [9,10,11].

A high neutrophil count is an indicator of activated non-specific inflammation, and lymphopenia is a marker of poor general health and physiological stress [12]. A high PLR and elevated platelet count can be useful in diagnosing systemic vasculitides [11]. However, it is unclear whether the NLR and PLR can be used as indicators to predict the development of acute symptoms of ARONJ. Therefore, we hypothesized that the systemic inflammatory status of patients with ARONJ can predict the development of acute symptoms of ARONJ. In this study, we investigated whether the NLR and PLR are associated with the development of acute symptoms in patients with ARONJ.

## 2. Materials and Methods

### 2.1. Patients

This retrospective, observational study included patients who had undergone medical examination at the Department of Oral and Maxillofacial Surgery, Mie University Hospital (Tsu, Mie). The survey was conducted using data obtained from patients diagnosed with ARONJ between 1 January 2011 and 31 December 2019. The inclusion criteria were as follows: age > 20 years, clinically confirmed ARONJ, and availability of complete medical records. The following data were extracted retrospectively from the hospital’s electronic medical records: age at diagnosis, sex, history of allergies, medical history, type of antiresorptive agent administered, medication period of antiresorptive agent, laboratory parameters at the time of diagnosis, NLR, and PLR. The NLR was calculated by dividing the absolute neutrophil count by the absolute lymphocyte count. The PLR was calculated by dividing the absolute platelet count by the absolute lymphocyte count. The following comorbid events were recorded: hospitalization for inflammation due to ARONJ and acute inflammation due to ARONJ. Acute inflammation in patients with ARONJ was defined as the exacerbation of symptoms, such as swelling, redness, pain, and drainage, and a C-reactive protein level of ≥0.3 mg/dL. The number of patients with acute inflammation onset (NAO) included those who experienced multiple episodes of acute inflammation onset due to ARONJ during this study. The period of stable condition (PSC) was defined as the period between the date of ARONJ diagnosis and the onset of acute inflammation. For cases without acute inflammation, PSC was defined as the period between the date of ARONJ diagnosis and the last follow-up. All patients were examined to grade the stage of ARONJ according to the criteria of the American Association of Oral and Maxillofacial Surgeons [1].

### 2.2. Statistical Analysis

The data are presented as the mean ± standard deviation (SD). All data were analyzed using R (version x64 4.1.1; R Foundation for Statistical Computing, Vienna, Austria) and RStudio (version 1.4.1717; RStudio, PBC, Boston, MA, USA). Continuous variables were analyzed using the Wilcoxon rank-sum test. Categorical variables were analyzed using Fisher’s exact test. Receiver operating characteristic (ROC) curve analysis was used to establish the cut-off values of the NLR and PLR. Multivariate logistic regression analysis was performed to assess patient characteristics. As a sub-analysis, we divided the patients into 3 groups according to the main disease (malignant tumor, osteoporosis, and autoimmune disease) and analyzed the NLR of each group by Kruskal–Wallis test and pairwise comparisons using the Wilcoxon rank sum test. All results were considered statistically significant at *p* < 0.05.

## 3. Results

### 3.1. Clinical Characteristics of Patients

Among the 169 patients diagnosed with ARONJ during the study period, 147 with available clinical records and laboratory data were included. The patient characteristics are presented in Table 1. Most patients were female (84, 57.1%), and the mean age was 73.2 years (standard deviation (SD), 10.7). Antiresorptive agents were administered for malignant tumors (85, 57.8%), osteoporosis (41, 27.9%), and autoimmune disease (21, 14.3%). Denosumab (62, 42.2%) and bisphosphonates (alendronate, risedronate, ibandronate, and minodronate; 85, 57.8%) were administered as antiresorptive agents. The mean NLR was 4.01 (SD, 4.37). In total, 49 (33.3%) patients progressed to acute inflammation and had an average PSC of 19.5 months (SD, 17.9 months). Furthermore, 33 patients (22.4 %) were hospitalized for acute inflammation due to ARONJ.

The associations between patient characteristics and the NLR are shown in Table 2. The high-NLR (≥4) and low-NLR (<4) groups had an average NLR of 4.01. There was a significant difference between the high- and low-NLR groups in terms of the white blood cell count (*p* < 0.001), neutrophil count (*p* < 0.001), lymphocyte count (*p* < 0.001), platelet count (*p* < 0.001), PLR (*p* < 0.001), hemoglobin level (*p* = 0.044), total protein level (*p* = 0.015), albumin level (*p* < 0.001), C-reactive protein level (*p* < 0.001), and hospitalization for inflammation (*p* = 0.033). In contrast, there were no significant differences between the groups in terms of sex (*p* = 0.372), age at diagnosis (*p* = 0.896), medical history (*p* = 0.095), history of allergies (*p* = 0.543), BMA use (*p* = 1.0), duration of BMA use (*p* = 0.897), NAO (*p* = 0.136), and PSC (*p* = 0.089).

### 3.2. ROC Curve Analysis for the Association between NLR and PLR and Hospitalization

ROC curve analysis between NLR and PLR and hospitalization for inflammation secondary to ARONJ showed that the cut off value is as follows. The NLR cutoff value was 2.833, with specificity and sensitivity of 0.579 and 0.758, respectively (Figure 1). The PLR cutoff value was 165.2, with specificity and sensitivity of 0.658 and 0.727, respectively, as shown in Figure 2.

### 3.3. Relationship between NLR and Hospitalization for Inflammation

Table 3 shows the characteristics of patients classified according to an NLR cutoff value of 2.833. Similar results to those in Table 2 are shown regarding the significant associations between the NLR and characteristic factors. There was a significant difference between the high (≥2.833) and low (<2.833) NLR groups in terms of the white blood cell count (*p* < 0.001), neutrophil level (*p* < 0.001), lymphocyte count (*p* < 0.001), platelet count (*p* < 0.001), PLR (*p* < 0.001), hemoglobin level (*p* = 0.013), total protein level (*p* = 0.033), albumin level (*p* = 0.0014), C-reactive protein level (*p* < 0.001), and hospitalization for inflammation (*p* < 0.001). In contrast, there were no significant differences between the groups in terms of sex (*p* = 0.868), age at diagnosis (*p* = 0.691), medical history (*p* = 0.054), history of allergies (*p* = 0.705), BMA use (*p* = 0.617), duration of BMA use (*p* = 0.617), NAO (*p* = 0.171), and PSC (*p* = 0.892).

Based on the PLR cut-off value, 63 and 84 patients were categorized into the high (≥165.2) and low (<165.2) PLR groups, respectively (Table 4). There was a significant difference between the high- and low-PLR groups in terms of neutrophil count (*p* = 0.007), lymphocyte count (*p* < 0.001), platelet count (*p* < 0.001), NLR (*p* < 0.001), hemoglobin level (*p* < 0.001), albumin level (*p* < 0.001), C-reactive protein level (*p* < 0.001), and hospitalization for inflammation (*p* < 0.001). However, there were no significant differences between the groups in terms of sex (*p* = 0.238), age at diagnosis (*p* = 0.372), medical history (*p* = 0.487), history of allergies (*p* = 1), BMA use (*p* = 1), period of BMA use (*p* = 0.721), white blood cell count (*p* = 0.813), total protein level (*p* = 0.052), NAO (*p* = 0.319), PSC (*p* = 0.352), and ARONJ stage (*p* = 0.433).

Sub-analysis according to the main disease (malignant tumor, osteoporosis, or autoimmune disease) revealed a significant difference between the NLR values of the osteoporosis and autoimmune disease groups (*p* = 0.028). There was no significant difference between the NLR of the malignant tumor group and that of the osteoporosis group (*p* = 1.0) or between the NLR of the malignant tumor group and that of the autoimmune disease group (*p* = 0.057).

Logistic regression analysis indicated that patients in the high-NLR group were more likely to be hospitalized for inflammation than those in the low-NLR group (*p* = 0.036; hazard ratio, 1.15; 95% confidence interval: 1.01–1.31). According logistic regression analysis, the albumin level, age, and ARONJ stage, these factors were found to not have a significant association with hospitalization for inflammation due to ARONJ (Table 5). Logistic regression analysis showed that patients in the high-PLR group had no significant association with hospitalization for inflammation compared with those in the low-PLR group.

## 4. Discussion

The role of the NLR and PLR in predicting acute inflammation in patients with ARONJ was investigated in this retrospective study. As expected, the patients with ARONJ were at a higher risk of developing acute inflammation. Therefore, independent prognostic factors for predicting acute inflammation in patients with ARONJ are essential. According to the study findings, the NLR and PLR were not significantly associated with acute inflammation. However, the NLR was significantly associated with hospitalization for ARONJ.

Considering the association between NLR and PLR and hospitalization for inflammation, we used the NLR and PLR as independent prognostic markers to predict hospitalization in patients with inflammation secondary to ARONJ.

The NLR and PLR are systemic markers of overall inflammation [13,14]. Inflammation, procoagulant imbalance, and endothelial dysfunction play important roles in the development of osteonecrosis of the jaw and its complications [15]. Inflammatory disorders often cause tissue damage, severe pain, and jaw dysfunction in patients with ARONJ [16,17] and often require hospitalization for inflammation of ARONJ.

As part of the treatment for the primary disease, patients receive BMAs. The occurrence of ARONJ as a side effect and the development of acute symptoms affect the treatment of the main disease and the QOL of the patient [18,19]. The identification of indicators that can predict the acuteness of ARONJ and the aggravation of symptoms is beneficial and can contribute to the treatment of major diseases and improve the QOL of patients. We believe that the PLR and NLR could be important predictive tools for the aggravation of symptoms in these patients.

Neutrophils are innate components of the immune response and play an important role in the inflammatory response. Bisphosphonates alter neutrophil levels, as evidenced by impaired polymorphonuclear leukocyte chemotaxis and nicotinamide adenine dinucleotide phosphate oxidase activity [20]. Further, neutrophil function is reduced in patients treated with bisphosphonates [21].

The NLR is a measurable parameter of systemic inflammation and has prognostic value in clinical conditions, such as inflammation, cardiovascular disease, and tumors [22]. The NLR is a biomarker that can be used to evaluate the inhibitory and excitatory activities of the immune system. Neutrophils can infiltrate the vascular wall and secrete superoxide radicals, cytokines, and various proteolytic enzymes that can cause endothelial damage, whereas lymphocytes can modulate the effect of neutrophils and have an anti-atherosclerotic role [10]. A high NLR indicates systematic inflammation due to higher neutrophilic activity, which can lead to worse outcomes [23,24].

In general, high platelet counts are associated with increased platelet activity [25]. Platelet activity may reflect an aggravated release of inflammatory mediators and promote destructive inflammatory processes [26,27]. High platelet counts represent increased thrombosis and the release of mediators that enhance inflammation. This may indicate an ongoing inflammatory condition and prothrombotic activity. Previous research findings suggest a bidirectional interaction between inflammation and thrombosis [14,28,29,30].

A limitation of this study was its retrospective design. Because of the single measurement in the diagnosis of ARONJ, there were various conditions of inflammation in each patient, and the changes in the NLR and PLR in response to treatment could not be estimated. Another limitation of this study was the lack of assessment of the correlation between the NLR or PLR and the condition of the main disease requiring BMA therapy. In future, we will consider increasing the number of target patients, exploring factors related to the NLR and PLR in predicting the onset and worsening of ARONJ symptoms, and prospectively investigating the association between the NLR or PLR and ARONJ symptoms. Future studies should aim to reduce the number of patients with secondary symptoms of ARONJ.

## 5. Conclusions

Our findings suggest that the NLR and PLR are not associated with acute inflammation in patients with ARONJ. However, the NLR has diagnostic value in predicting hospitalization for inflammatory events in patients with ARONJ.

## Figures and Tables

**Figure 1 diagnostics-12-01836-f001:**
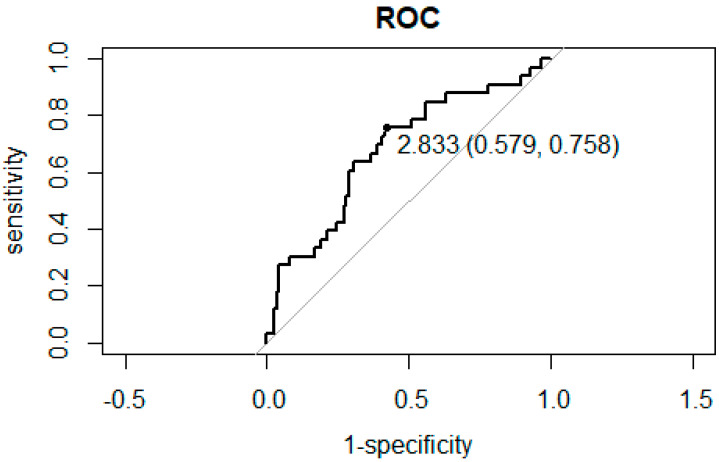
Receiver operating characteristic curve (ROC) analysis to obtain the sensitivity and specificity of NLR in predicting hospitalization for inflammation secondary to ARONJ. The dot indicates the cutoff value (sensitivity, specificity).

**Figure 2 diagnostics-12-01836-f002:**
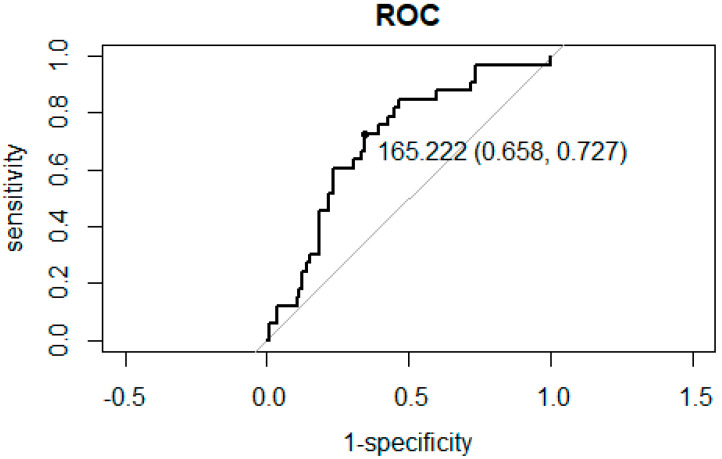
Receiver operating characteristic (ROC) curve analysis to obtain the sensitivity and specificity of PLR in predicting hospitalization for inflammation secondary to ARONJ. The dot indicates the cutoff value (sensitivity, specificity).

**Table 1 diagnostics-12-01836-t001:** Patient characteristics.

Patient Characteristics			*n* (%)	
Total			147	
Sex		Male	63 (42.9)	
		Female	84 (57.1)	
Main disease		Malignant tumor	85 (57.8)	
		Osteoporosis	41 (27.9)	
		Autoimmune disease	21 (14.3)	
Allergic history			37(25.1)	
BMA		Denosumab	62 (42.2)	
		Bisphosphonate related	85 (57.8)	
Hospitalization for acute inflammation of ARONJ			33 (22.4)	
NAO			49 (33.3)	
ARONJ stage (%)	0		10 (6.3)	
	1		45 (30.6)	
	2		86 (58.5)	
	3		6 (4.1)	
**Patient Characteristics**			**Mean (SD)**	
Age (years)			73.2 (10.7)	
Period of BMA use			43.9 (48.1)	months
Period of stable condition			19.5 (17.9)	months
Laboratory parameters at ARONJ diagnosis		White blood cells	6613.1 (2439.6)	×10^3^/μL
		Neutrophils	4456.3 (2362.4)	/μL
		Lymphocytes	1461.7 (601.9)	/μL
		Platelets	253.2 (97.9)	×10^3^/μL
		NLR	4.01 (4.37)	
		PLR	214.5 (158.9)	
		Hemoglobin	11.5 (1.76)	g/dL
		Total protein	6.92 (0.644)	g/dL
		Albumin	3.93 (0.452)	g/dL
		CRP	1.71 (5.144)	mg/dL

SD, standard deviation; CRP, C-reactive protein; BMA, bone-modifying agent; NLR, neutrophil–lymphocyte ratio; PLR, platelet–lymphocyte ratio; NAO, number of patients who had acute inflammation onset due to ARONJ; ARONJ, antiresorptive agent-related osteonecrosis of the jaw.

**Table 2 diagnostics-12-01836-t002:** Difference in patient characteristics between the low-NLR and high-NLR groups.

		Low NLR (<4)	High NLR (≥4)		*p*
*n*		100	47		
Sex (%)	Male	40 (40.0)	23 (48.9)		0.3722
Main disease (%)	Malignant tumor	61 (61.0)	24 (51.1)		
	Osteoporosis	29 (29.0)	12 (25.5)		
	Autoimmune disease	10 (10.0)	11 (23.4)		
Allergic history (%)		27 (27.0)	10 (21.3)		0.5434
BMA (%)	Denosumab	58 (58.0)	27 (57.4)		1
Hospitalization for acute inflammation of ARONJ (%)		17 (17.0)	16 (34.0)		0.0328 *
NAO (%)		18 (18.0)	15 (31.9)		0.08863
Age (mean (SD))		72.93 (11.24)	73.85 (9.38)		0.8958
Period of BMA use (mean (SD))		43.35 (46.39)	45.14 (52.28)	months	0.8966
Period of stable condition (mean (SD))		20.42 (19.31)	25.11 (21.10)	months	0.1363
ARONJ stage (mean (SD))		1.58 (0.68)	1.64 (0.67)		0.595
Laboratory parameters at ARONJ diagnosis (mean (SD))	White blood cells	5798.60 (1655.33)	8345.96 (2912.36)	×10^3^/μL	<0.001 *
	Neutrophils	3542.58 (1231.92)	6713.11 (2731.28)	/μL	<0.001 *
	Lymphocytes	1658.41 (576.95)	1043.04 (412.71)	/μL	<0.001 *
	Platelets	232.40 (79.12)	297.45 (118.37)	×10^3^/μL	<0.001 *
	NLR	2.29 (0.82)	7.68 (6.24)		<0.001 *
	PLR	156.47 (79.94)	337.98 (208.57)		<0.001 *
	Hemoglobin	11.72 (1.81)	11.16 (1.61)	g/dL	0.0441 *
	Total protein	7.01 (0.62)	6.74 (0.66)	g/dL	0.0154 *
	Albumin	4.03 (0.44)	3.72 (0.40)	g/dL	<0.001 *
	CRP	1.21 (5.94)	2.68 (2.90)	mg/dL	<0.001 *

SD, standard deviation; CRP, C-reactive protein; BMA, bone-modifying agent; NLR, neutrophil–lymphocyte ratio; PLR, platelet–lymphocyte ratio; NAO, number of patients with acute inflammation onset due to ARONJ; ARONJ, antiresorptive agent-related osteonecrosis of the jaw; * *p* < 0.05.

**Table 3 diagnostics-12-01836-t003:** Characteristics of patients grouped according to the NLR cutoff value.

		NLR < 2.833	NLR ≥ 2.833		*p*
*n*		74	73		
Sex (%)	Male	31 (41.9)	32 (43.8)		0.8683
Main disease (%)	Malignant tumor	43 (58.1)	42 (57.5)		0.054
	Osteoporosis	25 (33.8)	16 (21.9)		
	Autoimmune disease	6 (8.1)	15 (20.5)		
Allergic history (%)		20 (27.0)	17 (23.3)		0.7045
BMA (%)	Denosumab	41 (55.4)	44 (60.3)		0.6174
Hospitalization for acute inflammation of ARONJ (%)		8 (10.8)	25 (34.2)		<0.001 *
NAO (%)		13 (17.6)	20 (27.4)		0.1708
Age (mean (SD))		73.38 (11.28)	73.07 (10.06)		0.6911
Period of BMA use (mean (SD))		43.21 (50.09)	44.63 (46.19)	months	0.5536
Period of stable condition (mean (SD))		21.95 (20.29)	21.89 (19.73)	months	0.8921
The stage of ARONJ (mean (SD))		1.51(0.71)	1.68(0.64)		0.1612
Laboratory parameters at ARONJ diagnosis (mean (SD))	White blood cells	5747.70 (1598.59)	7490.27 (2815.55)	×10^3^/μL	<0.001 *
	Neutrophils	3320.57 (1129.17)	5808.93 (2619.85)	/μL	<0.001 *
	Lymphocytes	1790.15 (555.20)	1128.67 (445.91)	/μL	<0.001 *
	Platelets	229.38 (67.07)	277.34 (117.06)	×10^3^/μL	0.0128 *
	NLR	1.92 (0.59)	6.13 (5.41)		<0.001 *
	PLR	138.64 (55.58)	291.41 (190.18)		<0.001 *
	Hemoglobin	11.87 (1.80)	11.21 (1.67)	g/dL	0.0128 *
	Total protein	7.03 (0.63)	6.82 (0.64)	g/dL	0.0326 *
	Albumin	4.04 (0.45)	3.82 (0.43)	g/dL	0.00148 *
	CRP	1.48 (6.88)	1.94 (2.61)	mg/dL	<0.001 *

SD, standard deviation; CRP, C-reactive protein; BMA, bone-modifying agent; NLR, neutrophil–lymphocyte ratio; PLR, platelet–lymphocyte ratio; NAO, number of patients who had acute inflammation onset due to ARONJ; ARONJ, antiresorptive agent-related osteonecrosis of the jaw. * *p* < 0.05.

**Table 4 diagnostics-12-01836-t004:** Characteristics of patients grouped according to the PLR cutoff value.

		PLR < 165.2	PLR ≥ 165.2		*p*
*n*		84	63		
Sex (%)	Male	40 (47.6)	23 (36.5)		0.238
Main disease (%)	Malignant tumor	48 (57.1)	37 (58.7)		0.487
	Osteoporosis	26 (31.0)	15 (23.8)		
	Autoimmune disease	10 (11.9)	11 (17.5)		
Allergic history (%)		21 (25.0)	16 (25.4)		1
BMA (%)	Denosumab	49 (58.3)	36 (57.1)		1
Hospitalization for acute inflammation of ARONJ (%)		9 (10.7)	24 (38.1)		<0.001 *
NAO (%)		16 (19.0)	17 (27.0)		0.319
Age (mean (SD))		72.57 (11.08)	74.10 (10.09)		0.372
Period of BMA use (mean (SD))		42.61 (47.74)	45.68 (48.90)	months	0.721
Period of stable condition (mean (SD))		22.65 (19.17)	20.94 (21.05)	months	0.352
ARONJ stage (mean (SD))		1.56 (0.66)	1.65 (0.70)		0.433
Laboratory parameters at ARONJ diagnosis (mean (SD))	White blood cells	6504.64 (2075.96)	6757.62 (2864.99)	×10^3^/μL	0.813
	Neutrophils	4089.79 (1852.49)	5178.29 (2803.54)	/μL	0.007 *
	Lymphocytes	1783.02 (533.21)	1033.17 (382.21)	/μL	<0.001 *
	Platelets	212.21 (55.17)	307.84 (114.84)	×10^3^/μL	<0.001 *
	NLR	2.45 (1.21)	6.09 (5.94)		<0.001 *
	PLR	123.68 (28.05)	335.61 (179.83)		<0.001 *
	Hemoglobin	12.02 (1.56)	10.91 (1.83)	g/dL	<0.001 *
	Total protein	7.02 (0.56)	6.79 (0.73)	g/dL	0.052 *
	Albumin	4.08 (0.35)	3.73 (0.50)	g/dL	<0.001 *
	CRP	1.33 (6.49)	2.18 (2.74)	mg/dL	<0.001 *

* *p* < 0.05. SD, standard deviation; CRP, C-reactive protein; BMA, bone-modifying agent; NLR, neutrophil–lymphocyte ratio; PLR, platelet–lymphocyte ratio; NAO, number of patients who had acute inflammation onset due to ARONJ; ARONJ, antiresorptive agent-related osteonecrosis of the jaw.

**Table 5 diagnostics-12-01836-t005:** Logistic regression analyses of factors associated with hospitalization for inflammation of ARONJ.

**NLR**		
**Variables**	**HR (95%CI)**	** *p* **
Age	1.01 (0.973–1.06)	0.508
Albumin	0.642 (0.251–1.72)	0.359
ARONJ stage	1.59 (0.834–3.22)	0.172
NLR	1.15 (1.01–1.31)	0.036 *
**PLR**		
**Variables**	**HR (95%CI)**	** *p* **
Age	1.01 (0.973–1.06)	0.519
Albumin	0.665 (0.246–1.87)	0.422
ARONJ stage	1.42 (0.750–2.81)	0.296
PLR	1.00 (0.999–1.01)	0.151

* *p* < 0.05. HR, hazard ratio; CI, confidence interval; NLR, neutrophil–lymphocyte ratio; PLR, platelet–lymphocyte ratio; ARONJ, antiresorptive agent-related osteonecrosis of the jaw.

## Data Availability

The data presented in this study are available upon request from the corresponding author. The data are not publicly available because of the local guidelines for data storage.

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
