# Peer review of "Predictive Value of Neutrophil–Lymphocyte Ratio as a Marker in Antiresorptive Agent-Related Osteonecrosis of the Jaw: A Retrospective Analysis"

_diagnostics, 2022, doi:10.3390/diagnostics12081836_

Round 1
Reviewer 1 Report
This is a nicely designed clinical study driven to explore new diagnostic molecular tools for ARONJ. There are only minor comments for the discussion section. Please, see below:
The fourth paragraph of the discussion section is confusing. It is not clear if the authors are confronting their data with data from literature because there is no citation. If the statements are affirmative, without literature citation, it would sound speculative. I suggest to the authors to rewrite this section and clarify which references or data support the following statements: “As part of the treatment for the primary disease, patients receive BMAs. The occurrence of ARONJ as a side effect and the development of acute symptoms affect the treatment of the main disease and the QOL of the patient. Identification of indicators that can predict the acuteness of ARONJ and aggravation of symptoms is beneficial and can contribute to the treatment of major diseases and improve the QOL of patients. We believe that the PLR and can be important predictive tools for the aggravation of symptoms in these patients.” Please, try to review and improve this paragraph.
Line 244, 8th paragraph: “The NLR, but not PLR, predicted hospitalization for inflammatory events in patients 244 with ARONJ in this study.” Is this a conclusion? This paragraph is not linked to any other and seems to be disconnected from the sequence. Please, try to incorporate it within the discussion or in the conclusion section.
I believe that the present manuscript requires a minor review for clarity of the discussion section and conclusions.
Reviewer 2 Report
An interesting paper with precise scientific organization, accurate statistic analysis, and a significant number of cases.
As stated by the Authors themselves, some limits are related to the retrospective design of the study leading to aspected conclusions mostly already available in common knowledge. However, the study adds elements for a better comprehension of the evolution of the disease for each case.
The authors also show the intention to carry out further prospective studies to achieve more significant data and knowledge.
